## REVIEW ARTICLE

# The multiple mechanisms of MCL1 in the regulation of cell fate

Hayley Widden[1] & William J. Placzek [1 ✉]

MCL1 (myeloid cell leukemia-1) is a widely recognized pro-survival member of the Bcl-2 (B-cell lymphoma protein 2) family and a promising target for cancer therapy. While the role MCL1 plays in apoptosis is well defined, its participation in emerging non-apoptotic signaling pathways is only beginning to be appreciated. Here, we synthesize studies characterizing MCL1s influence on cell proliferation, DNA damage response, autophagy, calcium handling, and mitochondrial quality control to highlight the broader scope that MCL1 plays in cellular homeostasis regulation. Throughout this review, we discuss which pathways are likely to be impacted by emerging MCL1 inhibitors, as well as highlight non-cancerous disease states that could deploy Bcl-2 homology 3 (BH3)-mimetics in the future.

Myeloid cell leukemia-1 (MCL1) is a diverse cell regulatory protein, most notably recognized for its anti-apoptotic role in the Bcl-2 family. It was first identified in 1993 by Kozopas and colleagues in the laboratory of Ruth Craig while identifying genes that were associated with the transition from proliferation to differentiation in hematopoietic leukemia cells (ML-1)[1]. The novel gene was isolated during a phorbol-ester-induced differentiation experiment, which differentiates proliferating myeloblastic cells into non-proliferative monocytes or macrophages. The expression of *MCL1* was observed to increase just as the differentiation pathway was induced but before the appearance of canonical markers of differentiating cells or genes associated with the mature phenotype[1]. Soon after its identification, sequence homology of the *MCL1* gene with BCL2 and subsequent confirmation that MCL1 shared anti-apoptotic functionality solidified its position as a regulatory protein in intrinsic apoptosis[1–3].

MCL1 and BCL2 are anti-apoptotic members of a large class of proteins that modulate cell viability at the outer mitochondrial membrane (OMM). The Bcl-2 family consists of three subgroups: the anti-apoptotic members (MCL1, BCL2, BCLxL, BCLW, and BFL1/A1), the pore-forming effectors (BAK, BAX, and BOK), and the pro-apoptotic BH3-only proteins (e.g., NOXA, BIM, and PUMA)[4]. All Bcl-2 family proteins directly interact through protein–protein interactions to regulate intrinsic apoptosis by maintaining the integrity of the OMM. Upon irreparable cellular stress, the cell commits to an apoptotic signaling cascade through the upregulation of the pro-apoptotic BH3-only proteins and the downregulation or inactivation of pro-survival Bcl-2 family members. During this process, there is a shift in the Bcl-2 family interactome, which frees the effector proteins. Upon their release, effector proteins, most notably BAK and BAX, endure a confirmational change, promoting homo-oligomerization that forms cytotoxic pores in the OMM. These BAK and/or BAX oligomers release molecules such as cytochrome c into the cytosol, which initiates the rapid induction of intrinsic apoptosis[5–7]. The other two classes of the Bcl-2 family, the anti-apoptotic members, which include MCL1, and the pro-apoptotic BH3-only proteins, interact with both the pore-forming effectors and one another to modulate oligomerization and therefore cell survival[4].

The functional interplay between the Bcl-2 family subgroups surrounds one pivotal polypeptide motif that is shared by all members—the BH3 motif[8]. All members of the Bcl-2 family contain a BH3 motif, while some members contain additional BH motifs, BH1, BH2, and/or

[1]Department of Biochemistry and Molecular Genetics, University of Alabama at Birmingham, Birmingham, AL, USA. ✉email: placzek@uab.edu

BH4. The multi-motif members, which include all anti-apoptotic proteins (i.e., MCL1) and the pore-forming effectors, adopt a conserved globular fold with a hydrophobic binding pocket that mediates interactions with the amphipathic BH3 helix of the pro-apoptotic BH3-only proteins[5,9,10]. Therefore, the BH3-motif is the central mediator of intrinsic apoptotic regulation as it both promotes oligomerization of the pro-death proteins BAK and BAX and is utilized by the BH3-only proteins to bind and sequester their anti-apoptotic counterparts[4].

Significantly during tumorigenesis, the anti-apoptotic Bcl-2 family members are often upregulated through copy number alterations (e.g., MCL1) or are dysregulated as a result of chromosomal translocation (e.g., BCL2)[11,12]. This allows cancer cells to endure increased genotoxic stress that occurs during tumor development. To combat this vulnerability of cancer cell survival, small molecule inhibitors mimicking the regulatory BH3-motif were designed (BH3 mimetics)[13–17]. Prior to the development of BH3 mimetics, the primary focus of anti-apoptotic Bcl-2 family research efforts primarily revolved around how the anti-apoptotic proteins interact with canonical BH3 sequences. Yet, increasing recent evidence shows that MCL1 and the other Bcl-2 family members can bind to a diverse set of cell regulatory proteins through interactions using both its BH3-binding cleft and through distal sites in its protein structure. While recent reviews have discussed the non-apoptotic roles of BCL2 and BCLxL in detail[18], there has been a more limited discussion on the increasingly interesting body of research that directly connects MCL1 to the regulation of other cell signaling pathways beyond apoptosis. The literature summarized in this review synthesizes both studies that focus on the canonical BH3-binding cleft that may be impacted through BH3-mimetic targeting of MCL1, as well as other non-canonical protein interactions. Together, these studies demonstrate that MCL1 has many functions that alter cell fate, both as an anti-apoptotic Bcl-2 family protein and outside of that role through the regulation of cellular differentiation, cell cycle regulation, double-strand DNA break repair, mitochondrial dynamics, and bioenergetic metabolism.

## MCL1 is required for lineage-specific cell survival, differentiation, and maintenance

Healthy tissues must maintain homeostasis between cellular differentiation, proliferation, and death. At the nexus of these three pathways lies embryonic development, where differentiation, proliferation, and death are critically timed and regulated to ensure the proper development of the organism. MCL1 was initially recognized as an early marker for differentiation, yet the identification of sequence and structural homology between MCL1 and BCL2 led to the hallmark observation of their shared role in regulating intrinsic apoptosis[1]. Subsequent interrogation of MCL1-specific functions found that, unique from other Bcl-2 family members, MCL1 knockout mice exhibit peri-implantation lethality around E3.5[19]. During this developmental stage in mice, fertilized eggs are in the early blastocyst phase, which is comprised of two differentiated cell types: the inner cell mass and an outer layer of trophoblast cells. Of these two cell types, trophoblasts mature into the trophectoderm layer, which is required for implantation[20]. Recovered MCL1$^{-/-}$ blastocysts showed no gross morphological signs of apoptosis compared to the wild-type blastocysts and failed to hatch in vitro, suggesting a defect in trophoderm development[19]. These findings suggest that MCL1 is required for implantation during embryonic development and further established a role for MCL1 in cell biology that extends beyond apoptotic regulation early after its discovery.

Over the past two decades, MCL1 has proven to be essential for the survival and differentiation of many diverse cell types[21–24], most notably through hematopoiesis and neurogenesis[25–29]. Hematopoiesis is the process through which all blood cells are formed from self-renewing pluripotent stem cells that can differentiate into either myeloid or lymphoid cell types[30] (Fig. 1).

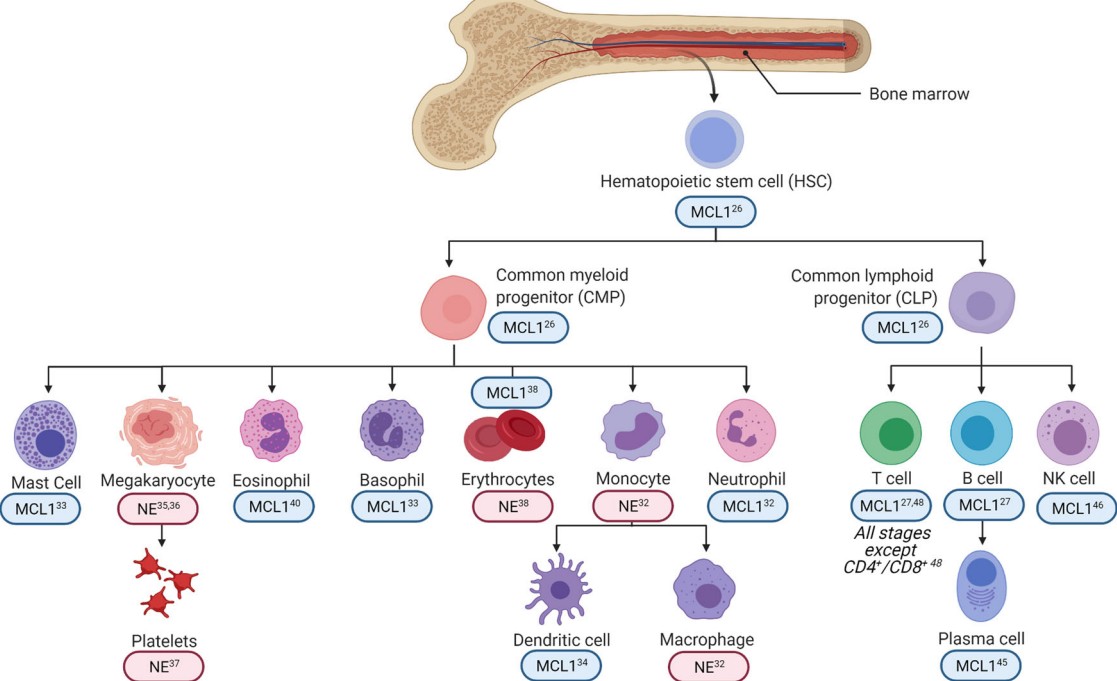

**Fig. 1 MCL1 differentially regulates hematopoietic cell survival and differentiation.** Stem cell differentiation from bone marrow. Blue boxes (MCL1) highlight cell types using in vivo modeling that solely require MCL1 expression for differentiation and/or survival. These cell types undergo a significant amount of apoptosis in vivo when MCL1 is selectively knocked out using genetic manipulation. Red boxes (NE) highlight cell types that exhibit no effect or a non-significant change in cell number compared to control mice when only MCL1 is knocked down.

The inducible deletion of *Mcl1* promotes hematopoietic failure through loss of hematopoietic stem cells, granulocyte monocyte progenitor cells, and common myeloid progenitor cells[2,26,27,31]. In addition to the survival and differentiation of myeloid progenitor cells[2], MCL1 is also required for myeloid-derived cell types such as neutrophils[32], mast cells[33], basophils[33], and dendritic cells[34]. In contrast, MCL1 is not solely responsible for the survival of monocytes[32], macrophages[32], megakaryocytes[35,36], or platelets[35,37] (Fig. 1). Furthermore, cellular dependence on MCL1 expression is influenced by precise timing and developmental stage. For example, MCL1 is required for early-stage erythropoiesis but is non-essential for maintenance of mature erythrocytes[38]. This selective requirement of MCL1 expression creates a unique vulnerability for specific immune cell targeting through MCL1 inhibition. Moreover, while not specifically responsible for macrophage cell survival in vivo, MCL1 modulates the macrophage effector response during bacterial phagocytosis, where the knockdown of MCL1 enhances sensitivity to macrophage cell death[32,39]. Phagocytes generally not only coordinate acute inflammation but also can cause inflammatory tissue damage through neutrophil recruitment or reduced eosinophil apoptosis[40]. Recently, the overexpression of MCL1 has been reported to exacerbate allergic airway inflammation and resist eosinophilic apoptosis[40]. To this end, targeting MCL1 in inflammatory lung conditions that induce acute respiratory distress or allergic airway disease, such as eosinophilic asthma, has been evaluated using the cyclin-dependent kinase (CDK) inhibitor AT7519[40–42]. These studies found that transcriptional silencing of MCL1 through CDK inhibitors promotes neutrophil apoptosis, while sparing macrophages and the phagocytosis of neutrophils[42]. In a separate study, AT7519 was shown to sensitize resistant eosinophils to apoptosis in mice transgenically overexpressing MCL1[40]. Significantly, these studies suggest a therapeutic opportunity for targeting MCL1 in inflammatory conditions as it promotes the resolution of neutrophils and eosinophils while sparing other lineages of the host immune system.

Just as MCL1 is required for hematopoietic stem cells and myeloid progenitor populations, it is also essential for common lymphoid progenitor cell survival and differentiation[26,27,31]. In B lymphocytes, MCL1 is required for the survival of germinal center B cells[43,44], memory B cells[44], and plasma cells[45]. Furthermore, MCL1 is required for natural killer cell survival[46] and throughout T cell development[27] (Fig. 1). MCL1 is essential for all stages of T cell maturation with the exception of early CD4+/CD8+ double positive thymocytes[47,48]. While not essential for double positive thymocyte survival, a conditional double knockout with BCLxL significantly decreases the survival of this T cell population[48]. Furthermore, MCL1 enhances the survival of memory CD8+ T cells after viral infection and promotes the formation of long-term lymphocyte memory[49]. In vivo modeling using an *MCL1* transgene demonstrated that T cells activated in response to viral infection shifted the population from short-lived effector cells to antigen-specific memory precursor cells. Memory precursor cells differentiate into the long-lived memory cell population, suggesting that MCL1 promotes long-term T cell memory formation[49]. Consistent with this study, MCL1 knockout mice with lymphocytic choriomeningitis viral infection exhibited decrease development of virus-specific T cells[50]. Unlike other examples where MCL1 and BCLxL are redundant, the overexpression of BCLxL did not rescue the effect of MCL1 knockdown[50]. Taken together, these studies highlight the broad role that MCL1 plays in hematopoietic processes ranging from cellular differentiation, effector function, and cell maintenance/death.

Paralleled with its broad role in hematopoietic cell survival and maturation, MCL1 is also critical in the tissue-specific differentiation and cellular survival in the central nervous system[28]. As shown through conditional knockout studies, MCL1 is required for neuronal development, where expression is high in neuronal precursor cells, promoting cellular survival throughout the process of neurogenesis[28]. Conditional Nestin-mediated *Mcl1* knockout mice, targeting both the neural stem cell and the intermediate neuronal progenitor cell population, are embryonic lethal at E16 with extensive cell death throughout the developing brain[28,51]. Moreover, MCL1 was recently identified as a critical survival factor in the midbrain dopaminergic neurons, whereas BCL2 and BCLxL promoted survival in non-dopaminergic cell lines[52]. Outside of the role in apoptotic regulation, MCL1 is required for neural progenitor cells to differentiate into post-mitotic neurons through the regulation of cell cycle exit and terminal mitosis. Following upregulation of MCL1, there is an increase in immature neurons that undergo terminal mitosis and premature differentiation, which are dependent on the modulation of cell cycle inhibitor p27[53]. As characterized by the studies described here, MCL1 plays a unique role in differentiation, both in hemopoietic lineages as well as in neurogenesis. BCL2 and BCLxL have independent and non-redundant roles compared with MCL1, many of which are reviewed elsewhere[18,29]. These studies together show that, rather than serving as redundant regulators of cellular apoptosis, timing and expression of the Bcl-2 family is essential for proper cell differentiation, cell proliferation, and cell death across multiple cell types. Complete understanding of these mechanisms and cell-specific functions are still being characterized with transgenic mouse studies and the utilization of emerging novel BH3 mimetics that allow for specific inhibition of anti-apoptotic Bcl-2 family members.

## MCL1 modulates cell division through interactions with cell cycle regulators

In addition to characterizing how MCL1 expression impacts specific cell lineages, it is also being interrogated for its impact on specific cell signaling pathways. MCL1 protein levels oscillate as cells progress through the cell cycle, with protein expression lowest in early G1 phase, increasing through S phase, and peaking in late S or early G2 phase[54]. Changes in MCL1 expression are a result of multiple levels of regulation from transcriptional activation, mRNA stability, non-coding RNA control, and post-translational modifications[55–58]. Specifically, during the cell cycle, MCL1 protein levels are regulated through two distinct phosphorylation sites on the MCL1 N-terminus, Ser64 and Thr92[55]. If there is a significant delay in mitotic progression (i.e., an unresolved problem in spindle assembly), Thr92 is phosphorylated by the mitosis-promoting CDK1-cyclin B1, which initiates the destruction of MCL1 through a proteasome-dependent mechanism[55]. This regulation of MCL1, in addition to the degradation by other E3 ligases such as FBW7 during mitosis, has led to its view as a mitotic clock that ensures timely exit from the cell cycle[59–61]. In cancer therapy, MCL1 degradation is an important event for apoptosis in cancer cells that are undergoing mitotic arrest by microtubule-stabilizing agents[55,59,61]. As the half-life and protein stability of MCL1 is regulated through interactions with its unique amino terminus[62], it is unsurprising that BCL2 and BCLxL do not exhibit similar changes in protein expression through the cell cycle[55]. This specific protein-level regulation makes the functionality of MCL1 in the cell cycle irreplaceable by other anti-apoptotic Bcl-2 family members[54].

Just as the precise regulation of MCL1 protein levels through the cell cycle ensures the successful completion of cell division through mitosis, there are also several MCL1 protein-binding partners that impact cell cycle progression at different phases. Utilizing its canonical BH3-binding cleft, MCL1 negatively regulates the INK4 cell cycle inhibitor, p18, to promote the G1/S transition and cell cycle entry in an Rb-dependent manner

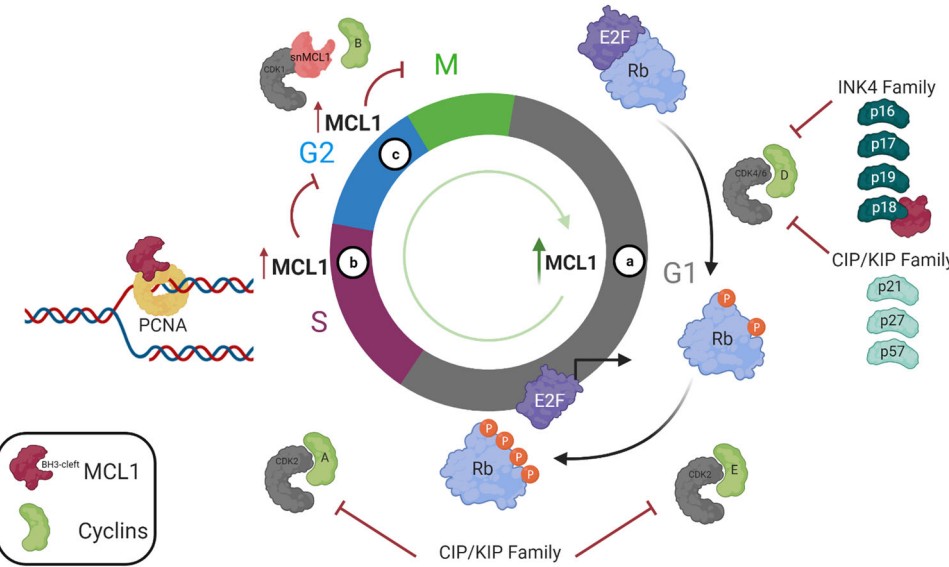

**Fig. 2 MCL1 modulates cell cycle entry and progression. a** In G1 phase, MCL1 binds and inhibits the stability of p18, an inhibitor of the G1/S transition. Decreased expression of p18 promotes cell cycle entry and proliferation in Rb-positive cells. **b** In S phase, MCL1 binds and inhibits PCNA, inhibiting DNA synthesis and cell cycle progression through G2. **c** In G2 phase, a nuclear isoform of MCL1 (i.e., snMCL1) binds to CDK1, which prevents the interaction with canonical binding partner, cyclin B. Ultimately, this function of snMCL1 inhibits mitotic completion.

(Fig. 2a). Mechanistically, MCL1 inhibits the stability of the p18 protein by targeting it for proteolytic degradation[63]. Another direct MCL1-binding partner in the cell cycle includes proliferating cell nuclear antigen (PCNA), a DNA sliding clamp that acts as a co-factor to DNA polymerase δ in S phase (Fig. 2b). In this specific study, overexpression of MCL1 inhibited cell cycle progression through S phase[64]. Mutational analysis of the PCNA-binding site (H224A, Glu225A, Thr226A, and Phe228A) demonstrated that mutant MCL1 still retains its anti-apoptotic function while promoting DNA synthesis. While the interaction with p18 involves a BH3-like interaction with MCL1s canonical BH3-binding cleft, the interaction with PCNA only involves eight amino acid residues within the hydrophobic p3 pocket of the MCL1s-binding interface[10,63,64]. It remains undetermined whether these two interactions are mutually exclusive or whether they solely depend on the phase of the cell cycle. Furthermore, it is still unsure whether these interactions will be impacted by MCL1-specific BH3 mimetics, although a recent study suggests that the cell cycle functions of MCL1 are spared[65].

MCL1's function in the cell cycle is not limited to its traditional full-length isoform. A proteolytically cleaved variant termed small nuclear MCL1 (snMCL1) was identified through western blot analysis and was exclusively localized in the nuclear cell fraction[66]. The 36 kD snMCL1 isoform exhibits direct binding to CDK1 and thereby inhibits the CDK1 interaction with its binding partner cyclin B (Fig. 2c). This complex is responsible for promoting MCL1 degradation in response to mitotic delay as discussed above[55]. Upon overexpression of snMCL1, the population of proliferating cells decreases consistent with a corresponding inhibition of the CDK1-cyclin B complex that promotes mitotic completion[66]. Through each of these interactions throughout the cell cycle, MCL1 overexpression has been demonstrated to promote cell proliferation or enable prolonged pausing in specific phases of the cell cycle, depending on the cell type, genetic background, and timing[53,63,64,66]. Further research is required to fully disentangle the mechanism by which MCL1 regulates cell proliferation and navigates cell cycle checkpoints. While many of the protein interactions detailed here are MCL1 specific, it should be noted that BCL2 and BCLxL have also been implicated in modulating cell cycle entry and progression through their own

unique mechanisms[67–69], which has been reviewed previously[70]. More in-depth investigation is required for both cancerous and normal cell types to determine how these interactions facilitate lineage-specific cell proliferation and the checkpoint response to intrinsic stress.

## MCL1 acts as a molecular switch for double-strand break (DSB) DNA repair

An important pathway intertwined with cell cycle regulation is DNA repair, which requires checkpoint activation and halting of cell proliferation. Two hallmarks of cancer that go hand in hand are genomic instability and enabling replicative immortality (i.e., cell proliferation)[71]. DNA damage response (DDR) is one of the most exploited mechanisms by which cancer cells simultaneously maintain viability and permit genomic mutation. To this end, it presents a promising focus for therapeutic intervention[72]. As regulators of apoptosis, the impact on DDR mediated by the pro-survival Bcl-2 family members including MCL1 has now been assessed in detail. The impact of MCL1 in DDR continues to evolve as it has been found to act not simply as an apoptotic switch but also as a key determinant of DDR activation and DSB repair. MCL1 was first observed to be upregulated in response to DNA damage induced by ionizing radiation, ultraviolet radiation, and alkylating agents in 1997[73]. This was the reciprocal effect observed for BCL2, whose expression decreases in response to DNA damage-induced therapy[73]. Likewise, expression of the pro-apoptotic effector protein, BAX, also increases following these treatments. Subsequent analysis found that, while BAX and other members of the Bcl-2 family are regulated by the tumor-suppressor p53[74], MCL1 is unique as it is upregulated through a p53-independent mechanism[73].

In response to DNA damage, MCL1 is upregulated, whereupon it binds and promotes the phosphorylation and subsequent activation of checkpoint kinase 1 (CHK1), a protein kinase responsible for halting the cell cycle for DNA repair[75]. Knock-down of MCL1 delays CHK1 phosphorylation, making this account significant as it was the first to identify that MCL1 plays a functional role in DDR[66,76] Furthermore, MCL1 binds to the early-response gene product-1, IEX-1, a protein that accumulates

at sites of DNA damage. Through this interaction, MCL1 and IEX-1 cooperate to maintain CHK1 activation with knockdown of either MCL1 or IEX-1 promoting genomic instability and resulting in increased sensitivity to replicative stress[77]. Ultimately, this could provide a novel area for combination therapy with MCL1 inhibition and DNA-damaging agents as a novel therapeutic strategy in cancer.

In addition to MCL1 facilitating the activation of DDR through CHK1, it also promotes specific mechanisms of DNA repair. The most lethal of all DNA lesions are DSBs that are resolved through one of the two mechanisms—homologous recombination (HR) or non-homologous end joining (NHEJ)[72]. HR is the more accurate mechanism, though it requires more time for repair and is cell cycle dependent as it requires a sister chromatid to serve as a repair template[78]. Alternatively, NHEJ is non-cell cycle dependent, but it is more error-prone as it ligates two ends of DNA back together, irrespective of templating[78]. Through several protein interactions between MCL1 and DDR machinery, increased steady-state MCL1 levels promote cells to repair DSBs using the more accurate HR mechanism over NHEJ. Specifically, at the sites of DNA damage, MCL1 binds to the phosphorylated minor histone, γH2AX[76], and co-localizes with 53BP1, post-irradiation[79] (Fig. 3a). Both γH2AX and 53BP1 are early signaling modifications at the sites of DNA damage, suggesting that MCL1 plays a direct role at the site of DSBs[76,79]. MCL1 knockdown delays γH2AX foci formation and promotes higher expression of residual 53BP1 and RIF1 foci, resulting in the accumulation of chromosomal abnormalities[75,76,79] (Fig. 3b). This accumulation of 53BP1-RIF1 foci promotes DNA end ligation required for NHEJ and attenuates DNA end-resection, a process required for repair through HR[78]. Moreover, MCL1 knockout cells exhibited reduced HR repair through a DSB reporter assay and decreased HR signaling protein foci (i.e., BRCA1, Rad51, and single-stranded RPA)[79], further suggesting that repair through HR has

been compromised (Fig. 3b). These studies of DSB foci formation demonstrate that MCL1 overexpression leads to a net increase in HR, while knockdown of MCL1 leads to an increase in repair through NHEJ. This suggests that MCL1 serves as a functional switch between HR and NHEJ in DSB repair[54,79].

The effect of MCL1 on DSBs goes beyond its effects on DDR protein foci accumulation. MCL1 binds and inhibits Ku, an essential NHEJ protein that stabilizes the DNA ends to promote ligation[54] (Fig. 3a). When MCL1 inhibits Ku and subsequently NHEJ in response to DSBs, it simultaneously promotes the HR machinery for more accurate DNA repair. DSBs repaired through HR require the assembly of three proteins, MRE11, RAD50, and NBS1, which are referred to as the MRN complex[78]. At the sites of DSBs, MCL1 facilitates the recruitment of MRE11[54] and binds to NBS1 in co-immunoprecipitation experiments[76]. The inhibitory action of MCL1 on Ku and the recruitment of the MRN complex further establishe MCL1 as a central regulator for the decision between DDR repair mechanisms. Like MCL1, other Bcl-2 family members have been identified in the DDR pathway including DSB repair. For example, BCL2 binds to MRE11, though unlike MCL1, inhibits the activity and promotes decreased HR repair[80]. More similarly, both MCL1 and BCL2 inhibit the Ku protein complex and subsequently inhibit DSB repair through NHEJ[81].

Characterizing these pathways in translational models provides insight into the benefit that cancer derives from the upregulation of MCL1, while also exposing possible combination therapies with emerging MCL1 inhibitors for cancer therapy. For instance, poly ADP-ribose polymerase (PARP) inhibitors (e.g., olaparib) promote DSBs through inhibition of the base excision repair pathway[82]. Typically, these lesions are repaired through HR, and if the cells are HR deficient (e.g., BRCA mutation), cells undergo apoptosis[82]. As knockdown of MCL1 drives cells from HR to the rapid, error-prone NHEJ, the combination of MCL1 and PARP

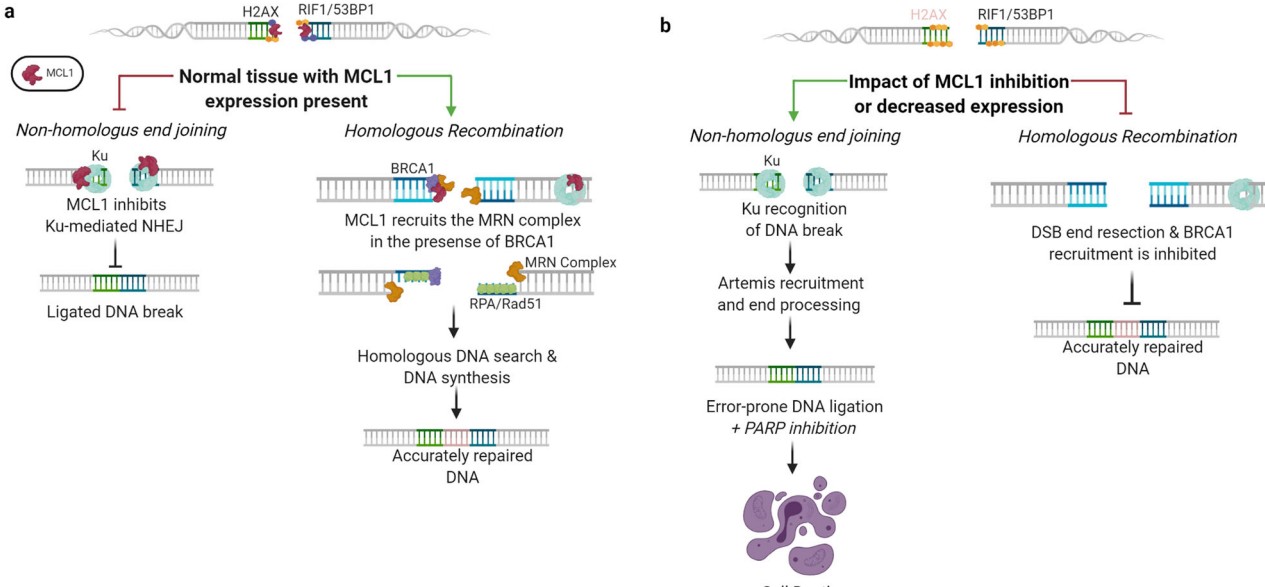

**Fig. 3 MCL1 facilitates double-strand break (DSB) DNA repair by stimulating homologous recombination (HR) and inhibiting error-prone non-homologous end joining (NHEJ). a** MCL1 expression in normal tissue. At the sites of DSBs, MCL1 co-localizes and facilitates γH2AX signaling foci, which promotes repair through HR. Furthermore, HR is activated by MCL1 through positive recruitment of the MRN complex, which promotes HR machinery such as BRCA1, ssRPA, and Rad51. MCL1 further inhibits NHEJ through the direct inhibition of the Ku protein complex. **b** MCL1 inhibition as a therapeutic opportunity for combination therapy in cancer. Upon MCL1 inhibition, HR repair is compromised as highlighted by reduced γH2AX foci at the sites of DBSs and decreased recruitment of BRCA1, ssRPA, and Rad51. 53BP1-RIF1 foci are increased, which promotes error-prone repair through NHEJ. NHEJ is further activated by decreasing the negative MCL1 regulation on Ku. While MCL1 inhibition promotes genomic instability, compromised HR opens a therapeutic window for synthetic lethality approaches that have previously been successful in cancer therapy.

inhibition was highlighted as a novel pathway to expose cancer cells to a targeted vulnerability[83]. To investigate whether MCL1 inhibition mimics the BRCA synthetic lethality approach, a recent study interrogated the combination of MCL1 knockdown with PARP inhibitors in a panel of xenograft models for colon, lung, and brain malignancies[83]. As MCL1 protein is rapidly turned over[62], MCL1 inhibition was achieved through mammalian target of rapamycin complex (mTORC) inhibition, which halts new protein synthesis[84]. By depleting MCL1 protein expression, cells are inherently primed to repair DSBs through NHEJ (Fig. 3b). Combination therapy using either of the mTORC inhibitors (everolimus or AXD2014) with PARP inhibition significantly inhibited cancer cell growth both in vitro and in vivo[83]. This is the first study to exploit MCL1-mediated vulnerabilities through a DNA repair mechanism versus the BH3-mimetic approaches that aim to sensitize cancer cells to intrinsic apoptosis. Significantly, targeting MCL1 through this mechanism did not induce apoptosis but rather resulted in activation of a necroptotic cell death pathway[83]. These findings pave the way for alternative approaches to therapeutic targeting of MCL1 while simultaneously exploiting its regulation of DSB DNA repair.

Outside of tumorigenesis and cancer maintenance, cells still require DDR mechanisms to repair endogenous lesions to DNA. One family of proteins that is required for proper execution of DNA repair is the p53 family of transcriptional tumor suppressors[85]. The p53 family is comprised of three homologous members (p53, p63, and p73) that are activated in response in DNA damage. Upon activation, the p53 family promotes the upregulation of target genes involved in DDR, halting the cell cycle, and precisely regulating apoptotic protein expression[85]. Interestingly, p63 and p73 contain a unique BH3-like sequence that specifically binds to the canonical BH3-binding cleft of MCL1[86,87]. The MCL1–p73 interaction inhibits the p73 DNA-binding capacity and transcriptional activation of DDR target genes[86]. Under normal circumstances, this mechanism could slow the induction of apoptotic target genes, such as BH3-only proteins NOXA and PUMA, to act as a time management sensor for repair before committing to the cell to apoptosis. Furthermore, under malignant conditions in which MCL1 is upregulated, the inhibition of MCL1 on p73 adds an additional layer of resistance on p73 activation. Specific inhibition of MCL1 using BH3-mimetic A-1210477 induces p73, which promotes cell cycle arrest, apoptosis, and DDR target gene expression, suggesting that MCL1 inhibition may reactivate p73 for more effective therapeutic response to platinum-based therapeutics[86]. This study provides yet another targeted protein interaction in the DDR pathway that could be exploited for combination therapy in cancer treatment regimens to activate DDR at the transcriptional level.

## MCL1 regulates autophagy and mitophagy through BH3-like protein interactions

Thus far, MCL1 has been described in cellular differentiation, cell cycle progression, and DDR, all of which are mechanisms of cell viability and repair. Outside of its canonical apoptotic role, MCL1 has also been characterized in other processes of cell death. Autophagy is a catabolic mechanism in which cellular constituents are degraded and recycled by the lysosome[88]. Activation of autophagy occurs in response to nutrient deprivation in which double-membrane vesicles called autophagosomes fuse with the lysosome for bulk degradation. Autophagosome nucleation is regulated by a protein complex that includes Beclin-1[88]. MCL1 and its anti-apoptotic homologs are able to directly bind and inhibit Beclin-1 through a consensus BH3 motif[89]. Inhibition of Beclin-1 by MCL1 subsequently mediates the balance between autophagic and apoptotic cell death[90,91]. In vitro binding studies have shown that both canonical BH3-only proteins and BH3 mimetics are able to disrupt the binding interaction between anti-apoptotic Bcl-2 family members and Beclin-1[92–94]. Importantly, MCL1 inactivation through phosphorylation or protein degradation is induced in response to nutrient deprivation allowing activation of the autophagic pathway[90,95]. This suggests that the inhibition of MCL1 can either induce apoptosis or autophagy, which depends on the parallel expression levels of Beclin-1 and other anti-apoptotic Bcl-2 family members that are also capable of binding and inhibiting Beclin-1-mediated autophagy[91].

In a more specialized narrative, MCL1 regulates mitophagy, which is the selective degradation of mitochondria via autophagy. Mitophagy is a critical process in mitochondrial maintenance and is regulated by both Beclin-1-independent and Beclin-1-dependent pathways[96]. Independent of Beclin-1, mitophagy is regulated through the Parkin/PINK1 pathway. Parkin is an E3 ligase that ubiquitinates depolarized mitochondria in a PINK1-dependent manner. Once tagged for degradation, damaged mitochondria have enhanced mitochondrial fission and decreased fusion to promote elimination by the autophagosome[96]. Here MCL1 and other anti-apoptotic Bcl-2 family members, with the exception of BCL-2 itself, inhibit mitophagy through the inhibition of Parkin translocation to depolarized mitochondria[97]. Furthermore, the selective elimination of damaged mitochondria was enhanced by BH3-mimetic ABT-737, which targets BCL2, BCLxL, and BCLW and UMI-77, which is an MCL1-specific inhibitor[97–99]. More recently, mitochondrial dysfunction and mitophagy pathways have a been evaluated as therapeutic targets in Alzheimer's Disease, highlighting a novel therapeutic area of BH3 mimetics in neurodegenerative diseases[98,99].

MCL1 also negatively regulates mitophagy through a Beclin-1-dependent pathway requiring pro-autophagic protein and mitophagy receptor, AMBRA1 and binding partner HUWE1. Like Beclin-1, both AMBRA1[100] and HUWE1 (i.e., Mcl-1 ubiquitin ligase E3 or MULE)[101] contain consensus BH3 motifs. When MCL1 is overexpressed, the recruitment of HUWE1 to the mitochondria is delayed, with a corresponding inhibition of mitochondrial ubiquitylation[102]. This finding was unique among the anti-apoptotic Bcl-2 family members, although an earlier publication by the same group identified an interaction between BCL2 and AMBRA1 as a crossroad between apoptosis and autophagic cell death pathways[100,102]. These studies suggest that MCL1 plays a role in mitophagy regulation through modulation of the AMBRA1 and PARKIN-mediated mitophagy pathways. Interestingly, a study published in 2014 suggested that the anti-apoptotic Bcl-2 family proteins such as MCL1 only indirectly modulate autophagic pathways through the canonical sequestration of BAK and BAX[103]. This claim is supported through knockdown experiments in which MCL1 has no impact on induction of autophagy in the absence of these effector proteins[103]. While there might be pathway and tissue-specific truth to these claims, the knockdown of Beclin-1, BAK, and BAX were all dispensable for Parkin/PINK1-dependent mitophagy[97]. The functional interplay between each of these pathways needs further investigation to fully elucidate the complete autophagic role of MCL1, which appears to be dependent on the expression of the other pro- and anti-apoptotic Bcl-2 family members.

## MCL1 modifies calcium homeostasis at the ER and mitochondrial membranes

All anti-apoptotic members of the Bcl-2 family, aside from BFL1/A1, contain a conserved C-terminal transmembrane domain that enables anchorage into the OMM[9] (Fig. 4a). As the transmembrane domain is not specific for the mitochondria, MCL1 also

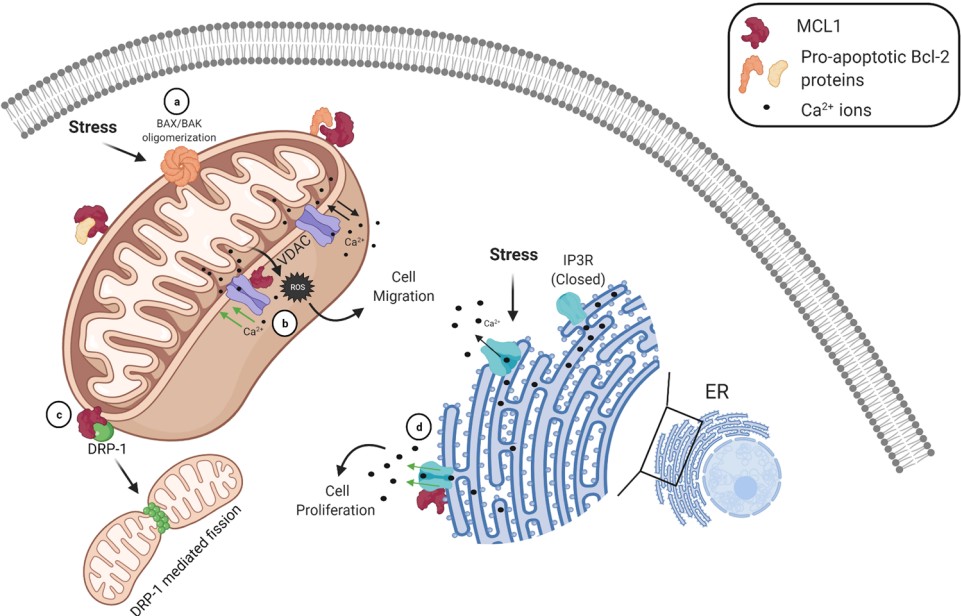

**Fig. 4 Membrane-bound MCL1 dysregulates calcium homeostasis and mitochondrial dynamics. a** MCL1 at the OMM binds to pro-apoptotic effector and BH3-only proteins (orange) to resist apoptosis. Upon irreparable cellular stress, MCL1 is downregulated, promoting BAX and/or BAK oligomerization as the rate-limiting step to the apoptotic cascade. **b** MCL1 binds to VDAC channels (purple) at the OMM, promoting mitochondrial $Ca^{2+}$ uptake, ROS production, and cell migration. **c** MCL1 binds to DRP-1 (green) at the OMM, modulating mitochondrial fission and fusion. **d** MCL1 localized to the ER membrane binds and impinges the activity of IP3R channels (blue), promoting basal $Ca^{2+}$ flux from the ER lumen into the cytosol, decreasing ER stores that are required for an array of cellular functions. The dysregulation on IP3R by MCL1 promotes unstimulated $Ca^{2+}$ release, activating cell proliferation. **b**, **d** Black arrows indicate the normal flux of calcium ions, whereas green arrows indicate the change in calcium release and uptake in response to MCL1 binding.

localizes to the endoplasmic reticulum (ER) and the mitochondria-associated ER membranes where it functions to regulate basal calcium ($Ca^{2+}$) flux[104]. Calcium release from the ER is an acute stimulus for both apoptotic and non-apoptotic cellular responses. In apoptosis, calcium is discharged from the ER lumen into the cytosol in response to cytochrome c release from depolarized mitochondria[105]. Furthermore, calcium ions are important biological signaling molecules in physiological responses like muscle contraction and neurotransmitter release[106]. Specifically in cancer cells, $Ca^{2+}$ signaling is required for cell proliferation, migration, and metastasis[107,108]. The Bcl-2 family proteins, BCL2 and BCLxL, have been studied and reviewed in detail describing their role in $Ca^{2+}$ regulation at the ER[104]. While some ER functions of BCL2 and BCLxL appear to be redundant with MCL1, others are not, providing an interesting area of unique Bcl-2 family functionality[104].

At the ER membrane, MCL1 binds to the carboxy-terminus of inositol triphosphate receptor (IP3R), which is directly responsible for $Ca^{2+}$ release from the ER lumen into the cytosol[109] (Fig. 4d). The IP3R channels regulate steady-state $Ca^{2+}$ concentrations through active uptake and passive release. Upon binding to IP3R, MCL1 increases the rate of pro-survival spontaneous $Ca^{2+}$ oscillations by impinging the gating activity of IP3R, promoting a lower concentration of ER-localized $Ca^{2+}$ stores[109]. While other anti-apoptotic Bcl-2 family members, such as BCLxL, decrease the expression of IP3R, MCL1 does not alter the expression of the IP3R receptor[110]. Furthermore, MCL1 has also been shown to bind and positively regulate all three isoforms of the mitochondrial $Ca^{2+}$ voltage-dependent anion channel (VDAC) at the mitochondria[111,112]. MCL1's interaction with VDAC promotes mitochondrial $Ca^{2+}$ uptake and stimulates reactive oxygen species (ROS) production, which promotes cell migration in non-small cell lung carcinoma cells[111] (Fig. 4b). Conversely, knockdown of MCL1 was shown to inhibit cell

migration without significantly impacting cell proliferation[111]. Collectively, MCL1 elicits pro-survival functions outside of its canonical regulation as a Bcl-2 family member, as it increases calcium oscillations from the ER and induces mitochondrial calcium uptake, which ultimately promotes cancer cell migration[104,109,111]. While other members of the Bcl-2 family bind to VDAC channels, MCL1 was favored over BCLxL in competitive pulldown experiments[111]. Furthermore, while the effect of older generation BH3 mimetics that target BCL2 and BCLxL have been explored[113–115], the impact that emerging MCL1 inhibitors have on $Ca^{2+}$ homeostasis remains unreported and should be an area of future investigation.

## MCL1 facilitates mitochondrial bioenergetics and dynamics

Thus far, we have demonstrated that MCL1 has critical homeostatic interactions with both BH3- and non-BH3-containing proteins in the nucleus, OMM, ER, and cytoplasm. MCL1 can also localize to the mitochondrial inner matrix (MIM) through a 33 amino acid mitochondrial matrix-targeting sequence in its N-terminus[116,117]. This sequence is proteolytically cleaved following translocation to the MIM resulting in a unique ΔN-MCL1 isoform[116]. Moreover, the MULE/Lasu1-binding site, which is one of the E3 ligases responsible for MCL1's rapid turnover, is removed, which reduces the steady-state ubiquitylation compared to the full-length isoform[118]. Initial identification of ΔN-MCL1 demonstrated conflicting evidence on the localization of the processed isoform, but MCL1 targeting to the MIM has now been confirmed by multiple studies[116–118]. While we will focus on MCL1 for the bulk of this section, both BCL2 and BCLxL have also been identified in the MIM[119–121]. BCLxL regulates bioenergetic metabolism through an interaction with the β-subunit of ATP synthase[119], while the localization and functionality of BCL2 in the MIM remains controversial[122].

While full-length MCL1 regulates intrinsic apoptosis and calcium flux at the OMM, ΔN-MCL1 regulates mitochondrial dynamics and bioenergetic metabolism in the MIM[117,123]. ΔN-MCL1, which is fully localized to the mitochondrial matrix, facilitates ATP production[123] and the maintenance of oligomeric ATP synthase[117]. MCL1 deletion in murine embryonic fibroblasts decreases the ability of complex I, complex II, and complex IV to transfer electrons through the electron transport chain[117]. Under these conditions, ATP production is reduced, which can be rescued with the expression of MCL1[117,123]. Furthermore, MCL1 deletion induced mitochondrial morphological changes, including loss of the wild-type tubular mitochondrial networks and the appearance of mitochondrial puncta, neither of which were present in the control cells[117]. In these MCL1 knockout cells, there was a delay in mitochondrial fusion that could also be rescued through the expression of wild-type MCL1 or ΔN-MCL1. An MCL1 transgene that only localizes to the OMM did not rescue the structural mitochondrial defects, demonstrating that ΔN-MCL1 is required for mitochondrial structure and bioenergetic metabolism[117]. Interestingly, ΔN-MCL1 does not bind and sequester pro-apoptotic BH3-only proteins[117].

In a more recent report published, the MCL1 BH3 helix was shown to positively regulate very long-chain acyl-CoA dehydrogenase (VLCAD) to promote fatty acid β-oxidation in the MIM[124]. Here knockdown of MCL1 or ΔN-MCL1 dysregulated long-chain fatty acid β-oxidation. Significantly, treatment with MCL1 small molecule inhibitor, S63845, had no impact on the rate of fatty acid β-oxidation of $^3$H-palmitic acid[124]. Importantly, this finding is consistent with the earlier studies demonstrating that ΔN-MCL1 does not bind to BH3-only proteins[117]. This result is supported by multiple other observations: (1) the interaction between MCL1 and VLCAD implies MCL1 is in an altered conformational state as the BH3 helix of MCL1 mediates binding and is therefore accessible[124], (2) loss-of-function mutations of the MCL1 hydrophobic groove do not alter ΔN-MCL1 mitochondrial functions[117], and (3) in vivo studies with S63845 showed minimal toxicity, suggesting that MIM-localized ΔN-MCL1 is spared from inhibition[17,117]. Upon analysis of patient data, MCL1 amplification was significantly correlated with a fatty acid β-oxidation gene signature in two independent acute myeloblastic leukemia datasets, suggesting a link between MCL1 upregulation and fatty acid β-oxidation metabolism in cancer[124]. To this end, another recent study found that upregulation of fatty acid β-oxidation pathways conferred resistance to Food and Drug Administration-approved BCL2-specific BH3-mimetic Venetoclax with Azacytidine treatment. In resistant leukemic stem cells, MCL1 inhibition did facilitate a decrease in fatty acid β-oxidation and confer sensitivity to the combination treatment with Azacytidine, prompting the need for further investigation into the crosstalk between the Bcl-2 family and fatty acid metabolism pathways[125]. Lastly, in breast cancer stem cells (CSCs), MCL1 is co-amplified with the MYC oncoprotein, which promotes oxidative phosphorylation that is independent of the canonical function in apoptotic regulation. This phenotype promotes an increase in ROS and accumulates hypoxia-inducible factor-α (HIF1α), which can be mediated by a pharmacological inhibitor of HIF1α but not by an MCL1 inhibitor alone[126]. This suggests that, while MCL1 inhibitors may be beneficial for some cancer cell types that rely on the BH3-binding pocket, resistant CSCs like those described here would be spared.

Paralleled with the functional consequence of MCL1 in mitochondrial bioenergetics such as cellular respiration, the generation of ATP, and metabolism, MCL1 also facilitates normal mitochondrial fission and fusion and regulates the ultrastructure of mitochondrial cristae[117]. MCL1 interacts with two guanosine triphosphatases (GTPases) that are responsible for remodeling the mitochondrial network—DRP-1[127] and OPA-1[128]. In mitochondrial dynamics, activated DRP1 forms oligomeric rings around the mitochondria at the OMM, dividing the organelle through mitochondrial fission[96]. Mitochondrial fission is a process that can mediate pro-survival or detrimental outcomes for the cell, especially in cardiac tissue where mitochondrial networks are extensive for energetic demands[129,130]. Conversely, OPA-1 mediates mitochondrial fusion at the MIM, the process by which two mitochondria physically merge into one[96]. MCL1 binds and promotes the stability of DRP-1, promoting mitochondrial fragmentation at the OMM[128] (Fig. 4c). On the contrary, ΔN-MCL1 inhibits OPA-1 protein stability at the mitochondrial matrix. While the stability of OPA-1 is decreased and fusion is repressed, MCL1 does not abolish its activity as mitochondria can still undergo fusion when MCL1 is inhibited[128]. Subsequent studies have since shown that MCL1-specific BH3 mimetics display fragmented mitochondrial networks, highlighting the shift in mitochondrial dynamics toward enhanced organelle fragmentation through fission[117,131,132] (Fig. 4c). MCL1 small molecule inhibitor, S63845, disrupted the mitochondrial network and MCL1's interaction with both DRP1 and OPA1, suggesting that the binding interface for these interactions is MCL1's hydrophobic BH3 cleft[128]. This is significant as it suggests that the use of MCL1 inhibitors will ultimately likely have an impact on the dynamic mitochondrial networks necessary for normal cellular homeostasis, contrary to earlier studies suggesting that this pathway would be spared.

## Concluding remarks: where are we now and where to go next?

As shown through the diverse functionality discussed here, MCL1 plays a critical role in cellular homeostasis, both through its canonical apoptotic and through emerging non-apoptotic roles. MCL1 is a highly studied protein; it has >2000 publications over the past 5 years with approximately 500 publications in 2020 alone. While a large portion of these studies, especially those since 2017, are characterizing MCL1-specific BH3 mimetics in various cancer cell types, it is clear that MCL1 plays a larger, understudied role in many aspects of cellular maintenance and stress response. While ongoing studies characterizing the impact of BH3 mimetics is essential to move these compounds from bench to bedside in disease therapy, characterizing the underlying biological role of MCL1 is essential to designing the best combination strategies with existing therapy. Furthermore, the fundamental biology will be critical to predict and understand the etiology of adverse events in clinical trials as these inhibitors are deployed in Phase I and II human clinical trials.

One anecdotal observation that can be gleaned from synthesizing this review is that MCL1 is a very diverse protein and many pathways must be monitored in deploying MCL1 inhibitors systemically. One tissue that relies heavily on mitochondrial metabolism and strict control over mitochondrial dynamics is the heart[133]. Cardiomyocytes must provide the energy required for circulation and oxygenation of organs throughout the body and mitochondria account for ~35% of cardiac tissue[133,134]. Comparatively to skeletal muscle, which is another high-energy-demanding tissue, mitochondria only comprise 3–8% skeletal muscle volume, which is highly dependent on physical activity[133]. Conditional knockout of MCL1 in cardiomyocytes in mice induces rapid cardiomyopathy and death[135,136]. Interestingly, these cells do not undergo apoptosis but have impaired autophagy and endure necrotic cell death[135]. Tissue analysis reveals that the MCL1$^{-/-}$ cardiomyocytes have swollen mitochondria, leading to the swelling and rupture of the OMM[135]. Mitochondria isolated from these cells have also reduced respiration, consistent with

reports of the roles of the essential role of MCL1 in the electron transport chain and ATP synthesis[117,135]. While MCL1-specific BH3 mimetics did not show significant toxicities in in vivo animal studies[17], there have been delays in ongoing Phase I/II clinical trials[137].

Outside characterizing emerging BH3 mimetics, there remains several areas of MCL1 biology that are underdeveloped as they have contradictory findings that suggest they are cell type and tissue specific. For example, MCL1s role in the cell cycle has not only been demonstrated to be proliferative but also delays populations in the S and G2 phase[63,64,66]. Furthermore, specifically in autophagic cell death, MCL1 deletion activates an autophagy response in murine cortical neurons[90], whereas MCL1 deletion in cardiomyocytes displayed autophagic dysfunction[135]. In many areas, BCL2 and BCLxL have been more extensively characterized, demonstrating both similar and unique roles between the two Bcl-2 family homologs[18,138]. It should be noted that, while many of these contradictory observations have been characterized using in vitro cell models, not all of these accounts can specifically discern a subsequent pro-survival impact from the canonical function of MCL1 versus a true non-canonical signaling mechanism. In the majority of the experiments described here, either BAK and/or BAX were present in the model systems and therefore cannot distinguish the difference between disrupting the balance of the Bcl-2 family or other novel intracellular signaling mechanisms. Thus, the role of MCL1 in these interactions requires further characterization to understand the role of MCL1 functionality in greater detail and the consideration of an MCL1/BAX/BAK knockout model should be considered for future studies. As BH3 mimetics are deployed in solid tumor models, it will be essential to determine whether the efficacy of these compounds relies on the apoptotic induction by BAK or BAX. For instance, a recent study was published highlighting that the dependance of breast cancer on MCL1 expression was solely dependent on its apoptotic function. In the absence of BAK and BAX, genetic deletion of MCL1 has no impact on breast cancer cells[139]. This study highlights the importance of the BAK/BAX double knockout cell lines to fully disentangle many of these apoptotic versus non-apoptotic pathways in MCL1 regulation.

Another new avenue of research in MCL1 biology encompasses a novel class of MCL1-binding proteins that contain a reverse BH3 (rBH3) motif, which specifically bind to MCL1 over the other anti-apoptotic Bcl-2 family members[87]. Recently, two rBH3-mediated interactions have been identified and characterized between cell cycle regulator p18[63] and tumor-suppressor p73[86]. Notably, while these proteins interact with MCL1 through the consensus BH3 pocket, they do not appear to impact MCL1s regulation of apoptosis at the OMM[63,86]. Instead, these novel protein interactions appear to impact their endogenous function in cell cycle regulation and transcriptional activation, respectively. The identification of the rBH3 has proven to be significant as these novel protein interactions are opening new avenues for non-canonical MCL1 protein regulation.

In conclusion, MCL1 is a dynamic, unique protein that is involved in a variety of cellular functions. These include but are not limited to cellular differentiation, cell cycle progression, DDR, autophagy, mitochondrial dynamics, calcium handling, and metabolism. MCL1, like BCL2, is also proving to be a valuable anticancer therapeutic target[15–17]. Thus, understanding the role of MCL1 across diverse cancer cell types and endogenous tissues is critical as MCL1 inhibitors progress through clinical trials. As we are currently in an era of personalized medicine, comprehending the immense role MCL1 executes in cellular physiology is critical. While the study of the Bcl-2 family over the past 30 years has focused on how signaling pathways transmit pro- and anti-apoptotic signals to regulate apoptosis, the emerging non-canonical functions of MCL1, BCL2, and BCLxL provide an improved model where these proteins can bind and promote many pro-survival and pro-proliferative signaling pathways. These feed into multiple central homeostatic regulatory systems that include but are not limited to transcription factor activation, cell cycle control, mitochondrial biogenesis, and metabolism. Therefore, while MCL1 has thus far been viewed simply as an apoptotic regulator, the multiple functions that these studies begin to explore suggest that it may be more significantly positioned as a rheostat that can report on cell stress on multiple levels that modulate ultimate cell fate decisions.

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

## Acknowledgements

This work was supported, in part, by funding from the National Institutes of Health Grant R01GM117391 (to W.J.P.) and the UAB Carmichael Scholarship (to H.W.). All figures were created using templates provided on BioRender.com. This content is solely the responsibility of the authors and does not necessarily represent the views of the National Institutes of Health. The funders had no role in the organization, interpretation, or writing of the manuscript or in the decision to publish this review.

## Author contributions

H.W. reviewed the literature; H.W. and W.J.P. wrote the manuscript.

## Competing interests

The authors declare no competing interests.
