## [Peer Review File · Communications Biology]

Reviewers' comments:

Reviewer #1 (Remarks to the Author):

This review provides a comprehensive survey of the literature, regarding MCL-1 apoptotic and non-apoptotic functions, with the main emphasis being on MCL-1 purported non-apoptotic functions. Review is well written/balanced, easy to read and supported well with good quality figures. A couple of points detailed below I think should be considered in a revised version.

- while many studies suggest non-apoptotic functions for MCL-1 (indeed BCL2 family proteins in general), in many cases controversy surrounds these studies due to the lack of important controls (e.g. do effects of targeting MCL-1 still occur in BAX/BAK deficient cells, are BH3-mimetic effects strictly defined as being on-target. For instance, some recently published work shows MCL-1 function in hematopoiesis and breast cancer being solely through its canonical function (see PMIDs 33512417, 33785871). While I'm not suggesting the authors critically appraise every cited article, I do think its important to discuss (perhaps in future directions) rigorous validation of non-canonical MCL-1 functions is often lacking, and propose generic approaches/controls to test for true non-canonical functions.

- case in point of the above, the early embryonic lethality of MCL-1 deficiency, may be due to a non-apoptotic function, as suggested by the authors "These findings suggest that MCL1 is required for implantation and further establishes a role for MCL1 in cell biology that extends beyond apoptotic regulation" - but in the absence of carrying out expts. in a BAX/BAK deficient background this is impossible to state in definitive terms

Reviewer #2 (Remarks to the Author):

In this review, the authors focus on non-apoptotic roles for the pro-survival BCL-2 family protein MCL-1. They provide a brief overview of MCL-1 and its widely appreciated role in mitochondrial/intrinsic apoptosis, and then cover more extensively its involvement in other cell signalling processes including development and cell differentiation, cell cycle progression, DNA repair, autophagy, calcium homeostasis, and mitochondrial metabolism. Whilst the importance of MCL-1 in controlling apoptosis is frequently reviewed, the perspective of focusing upon MCL-1 impacts on non-apoptotic signalling pathways is not often covered and will distinguish this review from others. I believe this review will be of interest to the field and have a few suggestions for improvement.

- Whilst the focus of the review is on MCL1 it may help to highlight (where appropriate) that some of the properties discussed have also been reported for related BCL-2-family pro-survival proteins so as not to give the impression they are unique to MCL1. The authors do point this out briefly when discussing calcium homeostasis and autophagy, but could add to their discussion in other parts of the review. For example, with respect to DNA repair processes, BCL2 itself has also been implicated in this process (e.g. Wang et al. Mol Cell. (2009) Bcl2 Negatively Regulates DNA Double-Strand-Break Repair through a Nonhomologous End-Joining Pathway. doi: 10.1016/j.molcel.2007.12.029; Xie et al. Nucleic Acids Research. (2015) Bcl2 inhibits recruitment of Mre11 complex to DNA double-strand breaks in response to high-linear energy. doi: 10.1093/nar/gku1358). Likewise, there is extensive literature on the involvement of BCL-2 pro-survival proteins in regulating the cell cycle (eg. O'Reilly et al. EMBO J. (1996) The cell death inhibitor BCL-2 and its homologues influence control of cell cycle entry. DOI: 10.1002/j.1460-2075.1996.tb01090.x) and mitochondrial energetics (e.g. Alavian et al. Nature Cell Biol. (2011) Bcl-xL regulates metabolic efficiency of neurons through interaction with the mitochondrial F1FO ATP synthase.).

- Some of the properties discussed intertwine with cell death and it can be difficult to ascertain whether the impact of MCL1 is truly through a non-canonical mechanism. A good control to provide confidence of this is evidence that the MCL1 impact persists even in the absence of BAX/BAK (e.g. must be separate from control of apoptosis). This type of control was included the

study of cardiac toxicity following MCL1 deletion in the Wang et al. Genes & Dev paper(ref 123). The result obtained with that control showed that BAX/BAK deletion improved cardiac function and animal survival following MCL1 deletion, suggesting that apoptosis was a substantial driver of the cardiomyopathy observed. However, because the phenotype was not completely reversed following BAX/BAK deletion and because mitochondrial abnormalities persisted in MCL1/BAX/BAK knockout heart tissue, it supported the hypothesis that MCL1 impacted mitochondrial function independently of its canonical role in apoptosis. It would be good to highlight where there is strong supportive evidence for the non-canonical roles discussed such as in this example, and others where similar evidence is available.

- There are a few recent or foundational papers that were overlooked and may fit well into some of the discussion points. The authors could consider incorporating some of these that cover the impact of MCL1 on:

Mitochondrial metabolism:

Lee et al. Cell Metabolism 2017. MYC and MCL1 Cooperatively Promote Chemotherapy-Resistant Breast Cancer Stem Cells via Regulation of Mitochondrial Oxidative Phosphorylation. DOI: 10.1016/j.cmet.2017.09.009

Autophagy:

Wakatsuki et al. J Cell Biol 2017. GSK3B-mediated phosphorylation of MCL1 regulates axonal autophagy to promote Wallerian degeneration. DOI: 10.1083/jcb.201606020.

Calcium Homeostasis:

Minagawa et al. J Biol Chem 2005. The anti-apoptotic protein Mcl-1 inhibits mitochondrial Ca²⁺ signals. DOI: 10.1074/jbc.M503210200.

A few technical points:

- line 40 suggests that cells commit to apoptosis through “upregulation” of BAX/BAK. This phrasing may be misleading. BAX/BAK are occasionally expressed at higher levels following stress, but this is generally not a major driver of commitment to apoptosis. Instead, their activation typically involves changes in their interactions with other BCL2 proteins that cause BAX/BAK to change conformation, dimerize/oligomerize and form pores – without changes to their expression level.

- lines 87-90. In my opinion it is not correct to state that MCL1 blastocysts showed no signs of apoptosis or to definitively rule out apoptosis as contributing to the early lethality of MCL1^{-/-} embryos based on the data in the Rinckenberger paper (ref 19).

-line 302. Remove “to”

-line 398. There are many E3 ligases that contribute to the rapid turnover of MCL1. Mule is one example, so it may be preferable to describe it as “one of the E3 ligases responsible”.

We would like to thank the reviewers for their careful reading and excellent suggestions to improve the quality of the review. We have gone through and made a number of changes to incorporate these suggestions and provide clear connection to prior reviews/publications that discuss the impact of BCL2 and BCLxL.

Reviewer #1 (Remarks to the Author):

This review provides a comprehensive survey of the literature, regarding MCL-1 apoptotic and non-apoptotic functions, with the main emphasis being on MCL-1 purported non-apoptotic functions. Review is well written/balanced, easy to read and supported well with good quality figures. A couple of points detailed below I think should be considered in a revised version.

- while many studies suggest non-apoptotic functions for MCL-1 (indeed BCL2 family proteins in general), in many cases controversy surrounds these studies due to the lack of important controls (e.g. do effects of targeting MCL-1 still occur in BAX/BAK deficient cells, are BH3-mimetic effects strictly defined as being on-target. For instance, some recently published work shows MCL-1 function in hematopoiesis and breast cancer being solely through its canonical function (see PMIDs 33512417, 33785871). While I'm not suggesting the authors critically appraise every cited article, I do think it's important to discuss (perhaps in future directions) rigorous validation of non-canonical MCL-1 functions is often lacking, and propose generic approaches/controls to test for true non-canonical functions.

We agree that additional analysis of these systems is necessary to fully determine how much of the impact of MCL1's function is fully independent of its interaction with BAK/BAX and have added a section, as recommended, that highlights the lack of experiments being performed in BAK/BAX knockout lines in the discussion (lines 516-522).

"It should be noted that while many of these contradictory observations have been characterized using in vitro cell models, not all of these accounts can specifically discern a subsequent pro-survival impact from the canonical function of MCL1 versus a true non-canonical signaling mechanism. In the majority of the experiments described here, either BAK and/or BAX were present in the model systems and therefore cannot distinguish the difference between disrupting the balance of the Bcl-2 family or other novel intracellular signaling mechanisms. Thus, the role of MCL1 in these interactions requires further characterization to understand the role of MCL1 functionality in greater detail and the consideration of an MCL1/BAX/BAK knockout model should be considered for future studies.. As BH3-mimetics are deployed in solid tumor models, it will be essential to determine if the efficacy of these compounds relies on the apoptotic induction by BAK or BAX. For instance, a recent study was published highlighting that the dependence of breast cancer on MCL1 expression was solely dependent on its apoptotic function. In the absence of BAK and BAX, genetic deletion of MCL1 has no impact on breast cancer cells¹³⁹. This study highlights the importance of the BAK/BAX double knockout cell lines to fully disentangle many of these apoptotic versus non-apoptotic pathways in MCL1 regulation."

- case in point of the above, the early embryonic lethality of MCL-1 deficiency, may be due to a non-apoptotic function, as suggested by the authors "These findings suggest that MCL1 is required for implantation and further establishes a role for MCL1 in cell biology that extends beyond apoptotic regulation" - but in the absence of carrying out expts. in a BAX/BAK deficient background this is impossible to state in definitive terms

We agree this was an overstatement and have modified it (lines 91-95).

"Recovered MCL1-/- blastocysts showed no gross morphological signs of apoptosis compared to the wild-type blastocysts and failed to hatch in vitro, suggesting a defect in trophectoderm development¹⁹."

Reviewer #2 (Remarks to the Author):

In this review, the authors focus on non-apoptotic roles for the pro-survival BCL-2 family protein MCL-1. They provide a brief overview of MCL-1 and its widely appreciated role in mitochondrial/intrinsic apoptosis, and then cover more extensively its involvement in other cell signalling processes including development and cell differentiation, cell cycle progression, DNA repair, autophagy, calcium homeostasis, and mitochondrial metabolism. Whilst the importance of MCL-1 in controlling apoptosis is frequently reviewed, the perspective of focusing upon MCL-1 impacts on non-apoptotic signalling pathways is not often covered and will distinguish this review from others. I believe this review will be of

interest to the field and have a few suggestions for improvement.

- Whilst the focus of the review is on MCL1 it may help to highlight (where appropriate) that some of the properties discussed have also been reported for related BCL-2-family pro-survival proteins so as not to give the impression they are unique to MCL1. The authors do point this out briefly when discussing calcium homeostasis and autophagy, but could add to their discussion in other parts of the review. For example, with respect to DNA repair processes, BCL2 itself has also been implicated in this process (e.g. Wang et al. Mol Cell. (2009) Bcl2 Negatively Regulates DNA Double-Strand-Break Repair through a Nonhomologous End-Joining Pathway. doi: 10.1016/j.molcel.2007.12.029; Xie et al. Nucleic Acids Research. (2015) Bcl2 inhibits recruitment of Mre11 complex to DNA double-strand breaks in response to high-linear energy. doi: 10.1093/nar/gku1358). Likewise, there is extensive literature on the involvement of BCL-2 pro-survival proteins in regulating the cell cycle (eg. O'Reilly et al. EMBO J. (1996) The cell death inhibitor BCL-2 and its homologues influence control of cell cycle entry. DOI: 10.1002/j.1460-2075.1996.tb01090.x) and mitochondrial energetics (e.g. Alavian et al. Nature Cell Biol. (2011) Bcl-xL regulates metabolic efficiency of neurons through interaction with the mitochondrial F1FO ATP synthase.).

We agree that BCL2 and BCLxL have significant impacts on the areas discussed. We chose to limit our analysis of these functions only due to space and as they have each recently been reviewed elsewhere. For each section, we have added additional text to ensure that the reader is clearly directed to these reviews and to make sure that readers are aware that MCL1 is not the sole anti-apoptotic Bcl-2 family member with non-apoptotic function.

Lines 213-216: *“While many of the protein interactions detailed here are MCL1-specific, it should be noted that BCL2 and BCLxL have also been implicated in modulating cell cycle entry and progression through their own unique mechanisms,⁶⁷⁻⁶⁹ which has been reviewed previously⁷⁰.”*

Lines 277-281: *“Like MCL1, other Bcl-2 family members have been identified in the DDR pathway including DSB repair. For example, BCL2 binds to MRE11, though unlike MCL1, inhibits the activity and promotes decreased HR repair⁸⁰. More similarly, both MCL1 and BCL2 inhibit the Ku protein complex and subsequently inhibit DSB repair through NHEJ⁸¹.”*

Lines 391-393: *“While other anti-apoptotic Bcl-2 family members, such as BCLxL, decrease the expression of IP3R, MCL1 does not alter the expression of the IP3R receptor¹¹⁰.”*

Lines 415-419: *“While we will focus on MCL1 for the bulk of this section, both BCL2 and BCLxL have also been identified in the MIM¹¹⁹⁻¹²¹. BCLxL regulates bioenergetic metabolism through an interaction with the β -subunit of ATP-synthase¹¹⁹, while the localization and functionality of BCL2 in the MIM remains controversial¹²².”*

- Some of the properties discussed intertwine with cell death and it can be difficult to ascertain whether the impact of MCL1 is truly through a non-canonical mechanism. A good control to provide confidence of this is evidence that the MCL1 impact persists even in the absence of BAX/BAK (e.g. must be separate from control of apoptosis). This type of control was included in the study of cardiac toxicity following MCL1 deletion in the Wang et al. Genes & Dev paper (ref 123). The result obtained with that control showed that BAX/BAK deletion improved cardiac function and animal survival following MCL1 deletion, suggesting that apoptosis was a substantial driver of the cardiomyopathy observed. However, because the phenotype was not completely reversed following BAX/BAK deletion and because mitochondrial abnormalities persisted in MCL1/BAX/BAK knockout heart tissue, it supported the hypothesis that MCL1 impacted mitochondrial function independently of its canonical role in apoptosis. It would be good to highlight where there is strong supportive evidence for the non-canonical roles discussed such as in this example, and others where similar evidence is available.

We agree that BAX/BAK deletion mutants are needed in the field and have added a large section in the discussion to highlight this (Lines 516-522).

- There are a few recent or foundational papers that were overlooked and may fit well into some of the discussion points. The authors could consider incorporating some of these that cover the impact of MCL1 on:

Mitochondrial metabolism:

Lee et al. Cell Metabolism 2017. MYC and MCL1 Cooperatively Promote Chemotherapy-Resistant Breast Cancer Stem Cells via Regulation of Mitochondrial Oxidative Phosphorylation. DOI: 10.1016/j.cmet.2017.09.009

Autophagy:

Wakatsuki et al. J Cell Biol 2017. GSK3B-mediated phosphorylation of MCL1 regulates axonal autophagy to promote Wallerian degeneration. DOI: 10.1083/jcb.201606020.

Calcium Homeostasis:

Minagawa et al. J Biol Chem 2005. The anti-apoptotic protein Mcl-1 inhibits mitochondrial Ca²⁺ signals. DOI: 10.1074/jbc.M503210200.

Thank you for the suggested references, we have incorporated all of these.

A few technical points:

- line 40 suggests that cells commit to apoptosis through “upregulation” of BAX/BAK. This phrasing may be misleading. BAX/BAK are occasionally expressed at higher levels following stress, but this is generally not a major driver of commitment to apoptosis. Instead, their activation typically involves changes in their interactions with other BCL2 proteins that cause BAX/BAK to change conformation, dimerize/oligomerize and form pores – without changes to their expression level.

- lines 87-90. In my opinion it is not correct to state that MCL1 blastocysts showed no signs of apoptosis or to definitively rule out apoptosis as contributing to the early lethality of MCL1^{-/-} embryos based on the data in the Rinckenberger paper (ref 19).

-line 302. Remove “to”

-line 398. There are many E3 ligases that contribute to the rapid turnover of MCL1. Mule is one example, so it may be preferable to describe it as “one of the E3 ligases responsible”.

Thank you for these suggestions, we have made the corrections.

REVIEWERS' COMMENTS:

Reviewer #1 (Remarks to the Author):

Authors have addressed all points that I raised.

Reviewer #2 (Remarks to the Author):

I am happy with the changes that have been made to the revised manuscript and have no further suggestions.